# A Family of Vertex Operator Algebras from Argyres-Douglas Theory

**Heeyeon Kim and Jaewon Song**

*Department of Physics, Korea Advanced Institute of Science and Technology*
*291 Daehak-ro, Yuseong-gu, Daejeon 34141, Republic of Korea*

*E-mail:* heeyeon.kim@kaist.ac.kr, jaewon.song@kaist.ac.kr

ABSTRACT: We find that multiple vertex operator algebras (VOAs) can arise from a single 4d $\mathcal{N} = 2$ superconformal field theory (SCFT). The connection is given by the BPS monodromy operator $M$, which is a wall-crossing invariant quantity that captures the BPS spectrum on the Coulomb branch. We find that the trace of the multiple powers of the monodromy operator $\text{Tr} M^N$ produces the vacuum characters of a VOA for each $N$. In particular, we realize unitary VOAs of the Deligne-Cvitanović exceptional series type $(A_2)_1$, $(G_2)_1$, $(D_4)_1$, $(F_4)_1$, $(E_6)_1$ from Argyres-Douglas theories. We also find the modular invariant characters of the 'intermediate vertex subalgebra' $(E_{7\frac{1}{2}})_1$ and $(X_1)_1$. Our analysis allows us to construct 3d $\mathcal{N} = 2$ gauge theories that flow to $\mathcal{N} = 4$ SCFTs in the IR, which gives rise to the topological field theories realizing the VOAs with these characters.

# 1   Introduction

Four-dimensional $\mathcal{N} = 2$ supersymmetric field theories have been known to have connections to two-dimensional field theories in multiple ways, where Seiberg-Witten geometry [1, 2] plays a vital role. One connection involves two distinctive objects: the spectrum of operators in a

conformal field theory and the spectrum of massive particles. It is a surprising connection since a conformal field theory does not have a particle state, and we only measure the critical exponents and correlators of primary operators.

The connection is given as follows: All $\mathcal{N} = 2$ supersymmetric theories are believed to have a Coulomb branch where low-energy dynamics is governed by $U(1)^r$ abelian gauge theory with massive charged particles. Such a particle spectrum can be determined from the Seiberg-Witten geometry, which exhibits wall-crossing phenomena as one moves in the Coulomb branch of the moduli space. Thanks to the extended supersymmetry, the protected part of the particle spectrum (BPS particle) is remarkably robust and sometimes can be determined precisely. On the other hand, at the origin of the moduli space (or at a conformal point), we do not have massive BPS particles, instead have a conformal field theory. The superconformal field theory has a protected sector, called the Schur sector, which can be mapped to a vertex operator algebra (VOA) [3]. It maps a unitary 4d SCFT $\mathcal{T}$ to a non-unitary VOA $\mathcal{V}$. Surprisingly, these two data - the spectrum of BPS particles and the spectrum of primary operators at the CFT point - turn out to be related in a precise way. Namely, the Schur index (defined as a limit of the superconformal index [4–6]) of the CFT point, which computes the vacuum character of the VOA can be written as [7]

$$I_S(q) = \chi_{\mathcal{V}}(q) = (q)_\infty^{2r} \operatorname{Tr} M(q)^{-1} \ , \tag{1.1}$$

where the operator $M(q)$ is the quantum monodromy operator in the quantum torus algebra [8], constructed out of the spectrum of the BPS operators at a given point in the Coulomb branch. Here $r$ is the rank (the dimension of the Coulomb branch) of the 4d theory, and $(q)_\infty \equiv \prod_{n \geq 0}(1 - q^n)$ is the q-Pochhammer symbol.

One of the most important features of the monodromy trace (1.1) is that it is a wall-crossing invariant quantity, and therefore it is independent of the choice of a Coulomb vacuum. A natural generalization of this formula is to consider the trace of other powers of the monodromy operator, which is by construction also a wall-crossing invariant. It was already observed in [8, 9] that the trace of this operator $\operatorname{Tr} M$ (without inverse) is given by the characters of various 2d vertex operator algebras. Higher powers of the monodromy operator, $\operatorname{Tr} M^N$ for $N > 1$ is considered in [9] (see also [10]) and it was conjectured that there is a family of associated VOAs $\{\mathcal{V}^{(N)}\}$ for a given 4d $\mathcal{N} = 2$ SCFT, whose vacuum character is computed by the formula $\operatorname{Tr} M^N$.

In the current paper, we find this is indeed the case by explicitly computing $\operatorname{Tr} M^N$ for a number of Argyres-Douglas theories with the central charge of the VOA $\mathcal{V}^{(N)}$ given by

$$c_{2d}^{(N)} = 12N c_{4d} - 2r(N + 1) \ . \tag{1.2}$$

This gives rise to the following schematic diagram:

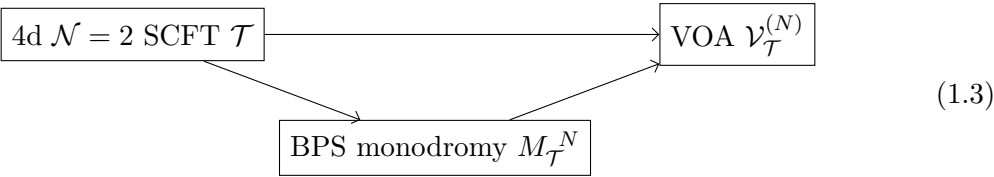

$$\text{(1.3)}$$

This correspondence extends the SCFT/VOA correspondence of [3] ($N = -1$ in our notation) to a family of VOAs labeled by $N$, which also includes *unitary* chiral algebras, from a 4d $\mathcal{N} = 2$ SCFT. For example, from the minimal $H_0 = (A_1, A_2)$ Argyres-Douglas theory [11], we obtain a series of vertex operator algebras ($N = 1, 2, 3, 4$):

$$(A_1, A_2): \quad \mathrm{osp}(1|2)_1, \ (G_2)_1, \ (F_4)_1, \ ``(E_{7\frac{1}{2}})_1" \ , \tag{1.4}$$

with central charges $c = \frac{2}{5}, \frac{14}{5}, \frac{26}{5}, \frac{38}{5}$ respectively. The $N = -1$ case is the previously well-known case of the Virasoro minimal model $M(2,5)$ with $c = -\frac{22}{5}$, and $N = 1$ gives the $\mathrm{osp}(1|2)_1$ algebra with $c = \frac{2}{5}$.[1] We notice that the VOAs in the above list are the ones that appear in the classification of RCFTs with two modules by Mathur-Mukhi-Sen (MMS) [15, 16]. Remarkably, we also find the modular invariant characters[2] of mysterious 'intermediate' VOA $(E_{7\frac{1}{2}})_1$ [17] from the minimal Argyres-Douglas theory! The $E_{7\frac{1}{2}}$ algebra appears as the missing hole in the Deligne-Cvitatonić exceptional series [18, 19]

$$A_1 \subset A_2 \subset G_2 \subset D_4 \subset F_4 \subset E_6 \subset E_7 \subset ``E_{7\frac{1}{2}}" \subset E_8 \ , \tag{1.5}$$

between $E_7$ and $E_8$ whose construction was given in [20]. The MMS classification precisely yields the affine Lie algebras of the above type with level 1 (except for the $(E_8)_1$, which has only one module).

In addition, by considering $(A_1, A_3)$, $(A_1, A_4)$ and $(A_1, D_4)$ theories [21–23], we obtain:

$$
\begin{aligned}
(A_1, A_3): &\quad (A_2)_1, \ (E_6)_1 \\
(A_1, A_4): &\quad \mathrm{osp}(1|4)_1, \ ``(X_1)_1" \\
(A_1, D_4): &\quad (D_4)_1
\end{aligned}
\tag{1.6}
$$

This means that we find all the MMS RCFTs from the simplest examples of Argyres-Douglas theories, except for the $(A_1)_1$ and $(E_7)_1$. We also find characters of another intermediate VOA, called as $(X_1)_1$ with $c = \frac{52}{7}$.

The physical meaning of the higher powers of monodromy operators can be most easily understood from the twisted circle compactification [8, 24–27].[3] Consider the holomorphic

---

[1]This is sometimes referred to as the conjugate Lee-Yang model or $M(5,2)_{\mathrm{eff}}$. The corresponding two modular functions are realized as the super-characters of two irreducible modules of simple affine VOA $\mathrm{osp}(1|2)$ at level one, whose central charge is $c = 2/5$ [12–14].

[2]Here, we use the standard terminology in math literature. By modular invariant characters, we mean that they form a basis of modular invariant subspace in the space of characters.

[3]Precursors to this correspondence was given in [28, 29].

topological twist of a 4d $\mathcal{N} = 2$ SCFT on $C_q \times \mathbb{C}$, where $C_q$ is a topologically twisted Melvin cigar. Compactification along the cigar circle involves a uniform $2\pi$ rotation of the $U(1)_r$ symmetry, which induces a Janus-like configuration along a closed loop in the Coulomb branch effective theory. The 3d BPS particles are then trapped at various loci on the loop according to their central charges, which gives rise to an effective 3d $\mathcal{N} = 2$ Chern-Simons matter theory description [25]. These theories are expected to flow to a superconformal theory with $\mathcal{N} = 4$ supersymmetry enhancement, which often admits a boundary condition that supports a rational VOA upon a topological twist [30–33]. This configuration is most interesting for non-Lagrangian SCFTs, where the Coulomb branch operators have fractional $U(1)_r$ charges, as in the Argyres-Douglas theories.

Furthermore, the BPS monodromy trace we study yields the characters in the so-called 'fermionic formulae' or the 'Nahm sum' form. Such form naturally appears as a half-index [34] of three-dimensional $\mathcal{N} = 2$ supersymmetric gauge theories with suitable boundary conditions [35], from which we can construct the corresponding 3d gauge theory [25].

The higher powers of the monodromy traces can be obtained by considering a twisted compactification along a closed path that winds around the Janus loop multiple times. It is natural to expect that there exists a positive integer $n$ for each theory where the $n$-th wrapping of the Janus loop trivializes the $U(1)_r$ twisting. For Argyres-Douglas theories, this number coincides with the least common multiple of the denominators of the R-charges of the Coulomb branch operators.

The multi-wrapping Janus-loop configurations give rise to a family of 3d $\mathcal{N} = 2$ Chern-Simons matter theories which are expected to flow to either an $\mathcal{N} = 4$ SCFT or a unitary TFT in the infrared. Alternatively, one can obtain such a description from the twisted reduction of a 4d $\mathcal{N} = 1$ Lagrangian description which flows to an $\mathcal{N} = 2$ Argyres-Douglas theory in the infrared [36–38], and considering the "higher sheets" of the partition function computation[4], as discussed in [26, 27, 29].

We can summarize the connections in the following diagram:

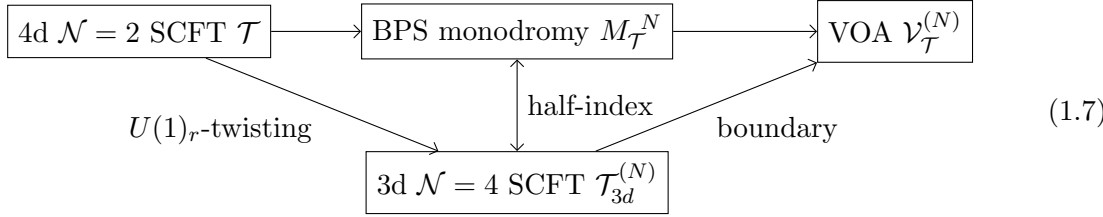

$$(1.7)$$

In this paper, we establish and explore this connection for a small sample of Argyres-Douglas theories.

The organization of this paper is as follows: In section 2, we spell out the details of the correspondence between monodromy trace and VOAs. In particular, we perform explicit computations for a series of Argyres-Douglas theories of low-rank. In section 3, we focus on

---

[4]The 'second sheet' of the superconformal index was considered in [39–41] to capture the asymptotic density of states in 4d SCFTs.

two particular cases that give rise to the modular invariant characters of intermediate VOAs $(E_{7\frac{1}{2}})_1$ and $(X_1)_1$ and construct the $\mathcal{N} = 2$ gauge theories that are expected to flow to 3d $\mathcal{N} = 4$ SCFT in the IR, which support VOAs whose vacuum characters coincides with these modular functions.

## 2  Vertex Operator Algebras from Monodromy Traces

In this section, we compute the monodromy traces for a number of Argyres-Douglas theories and match them with the characters of vertex operator algebras. Let $\Gamma$ be the electromagnetic charge lattice on the Coulomb branch with a pairing $\langle \, , \, \rangle$, which contains the flavor lattice $\Gamma_f$. The monodromy operator is defined as

$$M = \overset{\curvearrowright}{\prod_{\gamma}} \Psi_q(X_\gamma) \, , \tag{2.1}$$

where the product is over all BPS states of charge $\gamma \in \Gamma$ at a given chamber ordered according to the phase of central charges. The function $\Psi_q(X)$ is defined as

$$\Psi_q(X) = \prod_{n \geq 0}(1 + q^{\frac{1}{2}+n}X^n) = \sum_{k \geq 0}\frac{q^{\frac{k^2}{2}}}{(q)_k}X^k \, , \tag{2.2}$$

which can be thought of as a partition function of a BPS particle. Here we used the $q$-Pochhammer symbol

$$(z; q)_n \equiv \prod_{i=0}^{n-1}(1 - zq^i) \, , \quad (q)_n \equiv (q; q)_n \, , \quad (z; q) \equiv (z; q)_\infty \, . \tag{2.3}$$

The monodromy operator $M(q)$ is valued in the quantum torus algebra generated by non-commuting variables $X_\gamma$'s satisfying

$$X_{\gamma_1}X_{\gamma_2} = q^{\frac{1}{2}\langle \gamma_1, \gamma_2 \rangle}X_{\gamma_1+\gamma_2} = q^{\langle \gamma_1, \gamma_2 \rangle}X_{\gamma_2}X_{\gamma_1} \, . \tag{2.4}$$

We will often use the following identities:

$$\begin{aligned}
\Psi_q(X_{\gamma_1})\Psi_q(X_{\gamma_2}) &= \Psi_q(X_{\gamma_2})\Psi_q(X_{\gamma_1+\gamma_2})\Psi_q(X_{\gamma_2}) \\
\Theta(X_{\gamma_1})\Psi_q(X_{\gamma_2}) &= \Psi_q(X_{\gamma_1+\gamma_2})\Theta(X_{\gamma_1}) \\
\Theta(X_\gamma) &= \Psi_q(X_\gamma)\Psi_q(X_{-\gamma}) = \frac{1}{(q)_\infty}\sum_{n \in \mathbb{Z}}q^{\frac{n^2}{2}}X^n
\end{aligned} \tag{2.5}$$

These identities will turn out to be useful later.

The trace in the quantum torus algebra can be evaluated by imposing

$$\text{Tr}X_\gamma = \begin{cases} X_{\gamma_f} \, , & \text{if } \gamma = \gamma_f \in \Gamma_f \\ 0 \, , & \text{otherwise} \end{cases} \, . \tag{2.6}$$

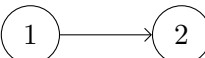

**Figure 1**: BPS quiver for the $(A_1, A_2)$ theory

Since the elements in $\Gamma_f$ are central, we can replace $X_{\gamma_f}$ by a complex number $x_f = \prod_i x_i^{f_i}$, where $f_i$ are flavor charges in some basis of $\Gamma_f$. This means that we impose the Gauss law constraint, taking contributions only from the gauge-invariant, neutral states. We impose a non-vanishing trace only if the corresponding BPS particle has neither electric nor magnetic charge.

Now let us consider the trace of a higher powers of the monodromy operator

$$I_N(q) \equiv (q)_\infty^{2r} \operatorname{Tr} M^N , \tag{2.7}$$

where $r$ is the rank (complex dimension of the Coulomb branch) of the underlying $\mathcal{N} = 2$ theory and $N$ is an integer. When this expression is convergent, we expect that it can be identified with the vacuum character of a vertex operator algebra $\mathcal{V}^{(N)}$.

We conjecture that the central charge $c_{2d}$ of $\mathcal{V}^{(N)}$ is given as

$$c_{2d}^{(N)} = 12Nc_{4d} - 2r(N+1) = N(12c_{4d} - 2r) - 2r , \tag{2.8}$$

where $c_{4d}$ is one of the central charges of the 4d $\mathcal{N} = 2$ SCFT and $r$ being the rank of the 4d theory. One way to see this relation is by invoking the SCFT/VOA correspondence of [3], which gives rise to the 2d central charge as $c_{2d} = -12c_{4d}$. This case corresponds to $N = -1$ in our setup, where the character of the VOA is given by the Schur index of the 4d SCFT. [7]

The higher powers of $M$ has been considered in [9] but with a different prefactor $(q)_\infty^{-2rN}$, giving the effective central charge $c_{2d}^{\text{CSVY}} = 12Nc_{4d}$. Taking the difference of the prefactor into account, we obtain the central charge (2.8). One can also directly compute the effective central charge by examining the asymptotic behavior of the trace as $q \to 1$. This gives the central charge for the $(G, G')$ Argyres-Douglas theory to be [9]

$$c_{2d}^{(N)} = \frac{Nr_G r_{G'} h_G h_{G'}}{h_G + h_{G'}} - 2r , \tag{2.9}$$

where $r_G$ and $h_G$ are the ranks and the dual coxeter numbers of $G$ respectivey and $r$ is the rank of the 4d SCFT.

In the remainder of this section, we test our proposals for a set of Argyres-Douglas theories of low-rank.

### 2.1 $(A_1, A_2)$ theory

The BPS quiver for the $(A_1, A_2)$ theory is given as in figure 1. The monodromy operator [8] for the $(A_1, A_2)$ theory is given as

$$M = \Psi_q(X_{\gamma_2})\Psi_q(X_{\gamma_1})\Psi_q(X_{-\gamma_2})\Psi_q(X_{-\gamma_1}) , \tag{2.10}$$

and it can be rewritten as follows:

$$M = \Theta(X_{\gamma_2})\Psi_q(X_{\gamma_1-\gamma_2})\Theta(X_{\gamma_1})$$
$$= \Theta(X_{\gamma_2})\Theta(X_{\gamma_1})\Psi_q(X_{-\gamma_2}) \tag{2.11}$$
$$= \Psi_q(X_{\gamma_1})\Theta(X_{\gamma_2})\Theta(X_{\gamma_1})$$

In this section, we evaluate the trace of (powers of) monodromy operator and match it with the character of certain vertex operator algebra. From (2.8) and (2.9), we expect the central charge for the corresponding VOA to be given as

$$c_{2d}^{(N)} = \frac{12N}{5} - 2 . \tag{2.12}$$

This gives the central charges for $N = 1, 2, 3, 4$ to be

$$c_{2d} = \frac{2}{5}, \frac{14}{5}, \frac{26}{5}, \frac{38}{5}. \tag{2.13}$$

As we will see, we find that the monodromy traces produce the characters for

$$\mathrm{osp}(1|2)_1, (G_2)_1, (F_4)_1, \text{``}(E_{7\frac{1}{2}})_1\text{''} , \tag{2.14}$$

respectively. Of course $N = -1$ case gives us the Lee-Yang minimal model $M(2,5)$ with $c = -\frac{22}{5}$. The meaning of the last entry in the above list will be explained momentarily.

### 2.1.1  Tr$M$: $\mathrm{osp}(1|2)_1$

The trace of the monodromy operator can be straightforwardly computed as

$$\mathrm{Tr}M = \mathrm{Tr}\,\Psi_q(X_{\gamma_2})\Psi_q(X_{\gamma_1})\Psi_q(X_{-\gamma_2})\Psi_q(X_{-\gamma_1})$$
$$= \mathrm{Tr}\,\Psi_q(X_{\gamma_1})\Psi_q(X_{\gamma_1+\gamma_2})\Psi_q(X_{\gamma_2})\Psi_q(X_{-\gamma_2})\Psi_q(-X_{\gamma_1})$$
$$= \mathrm{Tr}\,\Theta(X_{\gamma_1})\Psi_q(X_{\gamma_1+\gamma_2})\Theta(X_{\gamma_2})$$
$$= \frac{1}{(q)_\infty^2}\sum_{n_1,n_2\in\mathbb{Z}}\sum_{k\geq 0}\frac{q^{\frac{n_1^2+n_2^2+k^2}{2}}}{(q)_k}\mathrm{Tr}(X_{n_1\gamma_1}X_{k(\gamma_1+\gamma_2)}X_{n_2\gamma_2}) \tag{2.15}$$
$$= \frac{1}{(q)_\infty^2}\sum_{n_1,n_2\in\mathbb{Z}}\sum_{k\geq 0}\frac{q^{\frac{n_1^2+n_2^2}{2}}}{(q)_k}\mathrm{Tr}(X_{\gamma_1}^{n_1+k_1}X_{\gamma_2}^{n_2+k_2})$$
$$= \frac{1}{(q)_\infty^2}\sum_{k\geq 0}\frac{q^{k^2}}{(q)_k} .$$

From this, we obtain

$$I_1(q) = (q)_\infty^2\,\mathrm{Tr}M = \sum_{n\geq 0}\frac{q^{n^2}}{(q)_n} = \frac{1}{(q;q^5)(q^4;q^5)} , \tag{2.16}$$

where the second identity is the Rogers-Ramanujan identity. This gives the character of the non-vacuum module ($h = -\frac{1}{5}$) of the Lee-Yang minimal model (whose central charge is $c = -\frac{22}{5}$, with effective central charge $c_{\mathrm{eff}} = c - 24h_{\min} = \frac{2}{5}$) which is identical to the (specialized) vacuum character of the $\mathrm{osp}(1|2)_1$ whose central charge is $c = \frac{2}{5}$. This monodromy trace was already computed in [8, 9].

### 2.1.2 $\mathrm{Tr}M^2$: $(G_2)_1$

Now, let us consider the square of the monodromy operator. Its trace can be computed as

$$
\begin{aligned}
\mathrm{Tr}M^2 &= \mathrm{Tr}\,\Theta(X_{\gamma_1})\Psi_q(X_{\gamma_1+\gamma_2})\Theta(X_{\gamma_2})\Theta(X_{\gamma_1})\Psi_q(X_{\gamma_1+\gamma_2})\Theta(X_{\gamma_2}) \\
&= \frac{1}{(q)_\infty^4}\sum_{\boldsymbol{n},\boldsymbol{m}\in\mathbb{Z}^2}\sum_{k,l\geq0}\frac{q^{\frac{n_1^2+n_2^2+m_1^2+m_2^2}{2}}}{(q)_k(q)_l}\mathrm{Tr}(X_{\gamma_1}^{n_1+k}X_{\gamma_2}^{n_2+k}X_{\gamma_1}^{m_1+l}X_{\gamma_2}^{m_2+l}) \\
&= \frac{1}{(q)_\infty^4}\sum_{\boldsymbol{n},\boldsymbol{m}\in\mathbb{Z}^2}\sum_{k,l\geq0}\frac{q^{\frac{n_1^2+n_2^2+m_1^2+m_2^2}{2}}}{(q)_k(q)_l}q^{-(n_2+k)(m_1+l)}\mathrm{Tr}(X_{\gamma_1}^{n_1+k+m_1+l}X_{\gamma_2}^{n_2+k+m_2+l}) \\
&= \frac{1}{(q)_\infty^4}\sum_{\boldsymbol{n}\in\mathbb{Z}^2}\sum_{k,l\geq0}\frac{q^{2k^2+2kl+2kn_1+2kn_2+l^2+ln_1+ln_2+n_1^2+n_1n_2+n_2^2}}{(q)_k(q)_l} \,.
\end{aligned}
\tag{2.17}
$$

Upon multiplying with the prefactor, we obtain

$$
I_2(q) = (q)_\infty^2\,\mathrm{Tr}M^2 = 1 + 14q + 42q^2 + 140q^3 + 350q^4 + 840q^5 + 1827q^6 + \dots\,,
\tag{2.18}
$$

which agrees with the vacuum character of the $(G_2)_1$ to the leading orders in $q$-series expansion. Notice that $c = \frac{14}{5}$ we predicted above agrees with that of the Sugawara central charge of $(G_2)_1$. We conjecture that the last line of (2.17) gives the vacuum character of the $(G_2)_1$ affine Lie algebra.

### 2.1.3 $\mathrm{Tr}M^3$: $(F_4)_1$

The trace of the third power of the monodromy operator can be rewritten as

$$
\begin{aligned}
\mathrm{Tr}M^3 &= \mathrm{Tr}\,\Psi_q(X_{\gamma_1})[\Theta(X_{\gamma_2})\Theta(X_{\gamma_1})\Psi_q(X_{\gamma_1})][\Theta(X_{\gamma_2})\Theta(X_{\gamma_1})\Psi_q(X_{\gamma_1})]\Theta(X_{\gamma_2})\Theta(X_{\gamma_1}) \\
&= \mathrm{Tr}\,\Psi_q(X_{\gamma_1})\Psi_q(X_{\gamma_1+\gamma_2})[\Theta(X_{\gamma_2})\Theta(X_{\gamma_1})\Psi_q(X_{\gamma_1+\gamma_2})][\Theta(X_{\gamma_2})\Theta(X_{\gamma_1})]^2 \\
&= \mathrm{Tr}\,\Psi_q(X_{\gamma_1})\Psi_q(X_{\gamma_1+\gamma_2})\Psi_q(X_{\gamma_2})[\Theta(X_{\gamma_2})\Theta(X_{\gamma_1})]^3 \\
&= \mathrm{Tr}\,\Psi_q(X_{\gamma_2})\Psi_q(X_{\gamma_1})\,[\Theta(X_{\gamma_2})\Theta(X_{\gamma_1})]^3\,,
\end{aligned}
\tag{2.19}
$$

by applying the identities (2.5) and also (2.11). From this, we find

$$
\begin{aligned}
I_3(q) = (q)_\infty^2\,\mathrm{Tr}M^3 &= \frac{1}{(q)_\infty^4}\sum_{\boldsymbol{n}\in\mathbb{Z}^6}\sum_{k,l\geq0}\frac{q^{\frac{n_1^2+\dots+n_6^2+k^2+l^2}{2}}}{(q)_k(q)_l}\mathrm{Tr}(X_{\gamma_2}^l X_{\gamma_1}^k X_{\gamma_2}^{n_2}X_{\gamma_1}^{n_1}X_{\gamma_2}^{n_4}X_{\gamma_1}^{n_3}X_{\gamma_2}^{n_6}X_{\gamma_1}^{n_5}) \\
&= \frac{1}{(q)_\infty^4}\sum_{\boldsymbol{n}\in\mathbb{Z}^6}\sum_{k,l\geq0}\frac{q^{\frac{n_1^2+\dots+n_6^2+k^2+l^2}{2}+n_1n_4+n_3n_6+n_6n_1-lk}}{(q)_k(q)_l}\mathrm{Tr}(X_{\gamma_1}^{n_1+n_3+n_5+k}X_{\gamma_2}^{n_2+n_4+n_6+l}) \\
&= \frac{1}{(q)_\infty^4}\sum_{\boldsymbol{n}\in\mathbb{Z}^4}q^{n_1^2-n_1n_2+n_1n_3+n_2^2-n_2n_3+n_2n_4+n_3^2-n_3n_4+n_4^2}\sum_{k,l\geq0}\frac{q^{k^2-kl+kn_1+kn_3+l^2-ln_1+ln_2-ln_3+ln_4}}{(q)_k(q)_l} \\
&= 1 + 52q + 377q^2 + 1976q^3 + 7852q^4 + 27404q^5 + 84981q^6 + 243230q^7 + \cdots\,,
\end{aligned}
\tag{2.20}
$$

which agrees with the vacuum character of the $(F_4)_1$ to the leading orders in $q$-series expansion. The central charge $c = \frac{26}{5}$ agrees with the Sugawara central charge of $(F_4)_1$. We conjecture that this gives the vacuum character for the $(F_4)_1$ affine Lie algebra.

### 2.1.4  $\mathrm{Tr}M^4$: $(E_{7\frac{1}{2}})_1$

Using the identity (2.11), we can rewrite $\mathrm{Tr}M^4$ as

$$
\begin{aligned}
\mathrm{Tr}M^4 &= \mathrm{Tr}\ \Theta(X_{\gamma_2})\Theta(X_{\gamma_1})\Psi_q(X_{-\gamma_2})\left[\Psi_q(X_{\gamma_1})\Theta(X_{\gamma_2})\Theta(X_{\gamma_1})\right]^3 \\
&= \mathrm{Tr}\ \Theta(X_{\gamma_2})\Theta(X_{\gamma_1})\Psi_q(X_{-\gamma_2})\Psi_q(X_{\gamma_1})\Psi_q(X_{\gamma_1+\gamma_2})\Psi(X_{\gamma_2})\left[\Theta(X_{\gamma_2})\Theta(X_{\gamma_1})\right]^4 \quad (2.21) \\
&= \mathrm{Tr}\ \Psi(X_{\gamma_1})\left[\Theta(X_{\gamma_2})\Theta(X_{\gamma_1})\right]^4\Theta(X_{\gamma_2})\ .
\end{aligned}
$$

Expanding the last expression, we find

$$
\begin{aligned}
\mathrm{Tr}M^4 = \frac{1}{(q)_\infty^9}\sum_{n\geq 0}\frac{1}{(q)_n}\sum_{\{m_1,\cdots m_8,k\}\in\mathbb{Z}^9} & q^{\frac{1}{2}(n^2+k^2+m_1^2+\cdots m_8^2)}q^{-\frac{1}{2}[m_1(m_2+m_4+m_6+m_8)+m_2m_1]} \\
& \cdot q^{-\frac{1}{2}[m_3(m_4+m_6+m_8)+m_4(m_3+m_1)+m_5(m_6+m_8)]} \quad (2.22) \\
& \cdot q^{-\frac{1}{2}[m_6(m_5+m_3+m_1)+m_7m_8+m_8(m_7+m_5+m_3+m_1)]} \\
& \cdot \delta_{m_2+m_4+m_6+m_8+n,0}\cdot\delta_{m_1+m_3+m_5+m_7+k,0}\ \cdot
\end{aligned}
$$

Therefore

$$
I_4(q) = (q)_\infty^2\mathrm{Tr}M^4 = \frac{1}{(q)_\infty^7}\sum_{n\geq 0}\frac{1}{(q)_n}\sum_{\mathbf{m}\in\mathbb{Z}^7}q^{\frac{1}{2}\mathbf{m}^tK_8\mathbf{m}}\ , \quad (2.23)
$$

where $\mathbf{m} = (n,m_1,\cdots,m_7)$ and

$$
K_8 = \begin{pmatrix}
2 & 1 & 1 & 1 & 1 & 1 & 1 & 1 \\
1 & 2 & 0 & 1 & 0 & 1 & 0 & 1 \\
1 & 0 & 2 & 1 & 1 & 1 & 1 & 1 \\
1 & 1 & 1 & 2 & 0 & 1 & 0 & 1 \\
1 & 0 & 1 & 0 & 2 & 1 & 1 & 1 \\
1 & 1 & 1 & 1 & 1 & 2 & 0 & 1 \\
1 & 0 & 1 & 0 & 1 & 0 & 2 & 1 \\
1 & 1 & 1 & 1 & 1 & 1 & 1 & 2
\end{pmatrix} \quad (2.24)
$$

Note that the matrix $K_8$ defines a rank-8 unimodular even lattice, which must be isomorphic to the $E_8$ lattice. In fact, we can construct a $SL(8,\mathbb{Z})$ matrix $U$ such that $K_8 = UC(E_8)U^T$,

where

$$U = \begin{pmatrix} 1 & 0 & 0 & 0 & 0 & 0 & 0 & 0 \\ 0 & -1 & 0 & 0 & 0 & 0 & 0 & 0 \\ 0 & -1 & -2 & -2 & -2 & -1 & 0 & -1 \\ 0 & -1 & -1 & 0 & 0 & 0 & 0 & 0 \\ 0 & -1 & -2 & -3 & -4 & -2 & -1 & -2 \\ 0 & -1 & -1 & -1 & -2 & -1 & 0 & -1 \\ 0 & -1 & -2 & -3 & -3 & -2 & -1 & -2 \\ 0 & -1 & -1 & -1 & -1 & 0 & 0 & -1 \end{pmatrix} . \tag{2.25}$$

and $C(E_8)$ is the standard Cartan matrix of $E_8$:

$$C(E_8) = \begin{pmatrix} 2 & -1 & 0 & 0 & 0 & 0 & 0 & 0 \\ -1 & 2 & -1 & 0 & 0 & 0 & 0 & 0 \\ 0 & -1 & 2 & -1 & 0 & 0 & 0 & 0 \\ 0 & 0 & -1 & 2 & -1 & 0 & 0 & 0 \\ 0 & 0 & 0 & -1 & 2 & -1 & 0 & -1 \\ 0 & 0 & 0 & 0 & -1 & 2 & -1 & 0 \\ 0 & 0 & 0 & 0 & 0 & -1 & 2 & 0 \\ 0 & 0 & 0 & 0 & -1 & 0 & 0 & 2 \end{pmatrix} . \tag{2.26}$$

From this, we conclude that the change of basis from (2.24) to (2.26) leaves the first basis vector $\alpha_1$ invariant. This leads to the identity [5]

$$I_4(q) = (q)_\infty^2 \mathrm{Tr} M^4 = \frac{1}{(q)_\infty^7} \sum_{n \geq 0} \frac{1}{(q)_n} \sum_{\mathbf{m} \in \mathbb{Z}^7} q^{\frac{1}{2} \mathbf{m}^t K_8 \mathbf{m}}$$

$$= \frac{1}{(q)_\infty^7} \sum_{n_1 \geq 0} \sum_{\{n_2, \cdots, n_8\} \in \mathbb{Z}^7} \frac{q^{\frac{1}{2} \mathbf{n}^t C(E_8) \mathbf{n}}}{(q)_{n_1}} , \tag{2.27}$$

where $\mathbf{n} = (n_1, \cdots, n_8)$. The latter expression is nothing but one of the modular invariant characters of the intermediate vertex subalgebra $(E_{7\frac{1}{2}})_1$ [17], which has a $q$-expansion

$$I_4 = (q)_\infty^2 \mathrm{Tr}\ M^4 = 1 + 190q + 2831q^2 + 22306q^3 + \mathcal{O}(q^4) . \tag{2.28}$$

This is one of the solutions that appear in the MMS classification [16] of rational CFTs with two modules with $c = \frac{38}{5}$. This value agrees with the Sugawara central charge formula of the affine Lie algebra if we set $h^\vee = 24$ and $\dim(E_{7\frac{1}{2}}) = 190$. Combined with another character whose weight is $h = \frac{4}{5}$, we can construct a modular-invariant partition function with the putative chiral algebra. However, it was pointed out that [15] this leads to a negative fusion coefficient, so it cannot give a consistent RCFT. It is not known to us whether it is possible to

---

[5]We notice that the $q$-series in the first line converges much quicker compared to the one in the second line.

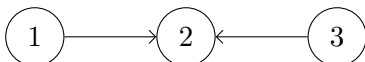

**Figure 2**: BPS quiver for the $(A_1, A_3)$ theory in the canonical chamber

have a rational VOA with a vacuum character given above. However, it is known that these two modular functions are realized as the characters of two irreducible modules of a rational and $C_2$-cofinite W-algebra $W_{-5}(E_8, f_\theta)$ in the *twisted sector* [42].[6] See also [43, 44].

## 2.2 $(A_1, A_3)$ theory

Let us now consider the simplest example with a flavor symmetry. The BPS quiver in the sink/source chamber for the $(A_1, A_3) = (A_1, D_3)$ theory is given as in figure 2. In this chamber, the monodromy operator can be written as

$$M = \Psi_q(X_{\gamma_1})\Psi_q(X_{\gamma_3})\Psi_q(X_{\gamma_2})\Psi_q(X_{-\gamma_1})\Psi_q(X_{-\gamma_3})\Psi_q(X_{-\gamma_2}) . \tag{2.29}$$

This theory has a flavor symmetry, and we can choose the flavor lattice vector as

$$\gamma_f = \gamma_3 - \gamma_1 , \tag{2.30}$$

so that

$$X_{\gamma_3} = X_{\gamma_1}X_{\gamma_f} . \tag{2.31}$$

The central charge for the VOA corresponding to $M^N$ is given as

$$c_{2d}^{(N)} = 4N - 2 , \tag{2.32}$$

so that we get $c = 2$ for $N = 1$ and $c = 6$ for $N = 2$. We argue that they corresponds to $(A_2)_1$ and $(E_6)_1$ respectively.

### 2.2.1 Tr$M$: $(A_2)_1$

Upon using the identities we introduced above, we find

$$\begin{aligned}
\mathrm{Tr}M &= \mathrm{Tr}\ \Psi_q(X_{\gamma_1})\Psi_q(X_{\gamma_3})\Psi_q(X_{\gamma_2})\Psi_q(X_{-\gamma_1})\Psi_q(X_{-\gamma_3})\Psi_q(X_{-\gamma_2}) \\
&= \mathrm{Tr}\ \Psi_q(X_{\gamma_1})\Psi_q(X_{\gamma_2})\Theta(X_{\gamma_3})\Theta(X_{\gamma_2})\Psi_q(X_{-\gamma_1-\gamma_2})\Psi_q(X_{-\gamma_1}) \\
&= \mathrm{Tr}\ \Theta(X_{\gamma_1})\Theta(X_{\gamma_2})\Theta(X_{\gamma_3})\Theta(X_{\gamma_2}) .
\end{aligned} \tag{2.33}$$

From here, we find

$$\begin{aligned}
I_1(q) = (q)_\infty^2 \mathrm{Tr}M &= \frac{1}{(q)_\infty^2} \sum_{\boldsymbol{n} \in \mathbb{Z}^4} q^{\frac{1}{2}(n_1^2+n_2^2+n_3^2+n_4^2)}\mathrm{Tr}(X_1^{n_1}X_2^{n_2}X_3^{n_3}X_2^{n_4}) \\
&= \frac{1}{(q)_\infty^2} \sum_{\boldsymbol{n} \in \mathbb{Z}^2} q^{\frac{1}{2}\boldsymbol{n}^T C(A_2)\boldsymbol{n}} \\
&= 1 + 8q + 17q^2 + 46q^3 + 98q^4 + 198q^5 + 371q^6 + \cdots ,
\end{aligned} \tag{2.34}$$

---

[6]We thank T. Arakawa for pointing out this to us.

where we have used $X_3 = X_1 X_f$ with $\text{Tr} X_f = 1$ and $C(A_2)$ is the Cartan matrix for the Lie algebra $A_2$. This is the theta function for the $A_2$ root lattice and, indeed, is identical to the vacuum character of the affine Lie algebra $(A_2)_1$.

### 2.2.2  $\text{Tr} M^2$: $(E_6)_1$

Upon evaluating the trace of the square of the monodromy operator, we obtain

$$
\begin{aligned}
\text{Tr} M^2 &= \text{Tr}\ \Psi_q(X_{\gamma_1})\Psi_q(X_{\gamma_2})\Theta(X_{\gamma_3})\Theta(X_{\gamma_2})\Psi_q(X_{-\gamma_1-\gamma_2})\Psi_q(X_{-\gamma_1})\Psi_q(X_{\gamma_1}) \\
&\quad \times \Psi_q(X_{\gamma_2})\Theta(X_{\gamma_3})\Theta(X_{\gamma_2})\Psi_q(X_{-\gamma_1-\gamma_2})\Psi_q(X_{-\gamma_1}) \\
&= \text{Tr}\ \Theta(X_{\gamma_1})\Theta(X_{\gamma_2})\Theta(X_{\gamma_3})\Theta(X_{\gamma_2})\Theta(X_{\gamma_1})\Theta(X_{\gamma_2})\Theta(X_{\gamma_3})\Theta(X_{\gamma_2})\ .
\end{aligned}
\tag{2.35}
$$

Therefore, we get

$$
\begin{aligned}
I_2(q) &= (q)_\infty^2 \text{Tr} M^2 = \frac{1}{(q)_\infty^6} \sum_{\boldsymbol{n}\in\mathbb{Z}^8} q^{\frac{1}{2}\boldsymbol{n}\cdot\boldsymbol{n}}\ \text{Tr}(X_1^{n_1} X_2^{n_2} X_3^{n_3} X_2^{n_4} X_1^{n_5} X_2^{n_6} X_3^{n_7} X_2^{n_8}) \\
&= \frac{1}{(q)_\infty^6} \sum_{\boldsymbol{n}\in\mathbb{Z}^8} q^{\frac{1}{2}\boldsymbol{n}\cdot\boldsymbol{n}-n_2 n_3 + n_5 n_6 - (n_5+n_7)(n_2+n_4+n_6)}\ \delta_{n_1+n_3+n_5+n_7,0}\cdot\delta_{n_2+n_4+n_6+n_8,0} \quad (2.36) \\
&= \frac{1}{(q)_\infty^6} \sum_{\boldsymbol{n}\in\mathbb{Z}^6} q^{\boldsymbol{n}^T K(E_6)\boldsymbol{n}}
\end{aligned}
$$

where

$$
K(E_6) = \begin{pmatrix}
2 & 1 & 1 & 1 & 1 & 1 \\
1 & 2 & 0 & 1 & 0 & 1 \\
1 & 0 & 2 & 1 & 1 & 1 \\
1 & 1 & 1 & 2 & 0 & 1 \\
1 & 0 & 1 & 0 & 2 & 1 \\
1 & 1 & 1 & 1 & 1 & 2
\end{pmatrix}.
\tag{2.37}
$$

Again we find that $K(E_6) = U C(E_6) U^T$ with a $GL(6,\mathbb{Z})$ matrix

$$
U = \begin{pmatrix}
1 & 0 & 0 & 0 & 0 & 0 \\
0 & -1 & 0 & 0 & 0 & 0 \\
0 & -1 & -2 & -1 & 0 & -1 \\
0 & -1 & -1 & -1 & 0 & -1 \\
1 & 1 & 1 & 1 & 1 & 1 \\
0 & -1 & -1 & -1 & 0 & 0
\end{pmatrix}.
\tag{2.38}
$$

This allows us to write

$$
\begin{aligned}
I_2(q) &= \frac{1}{(q)_\infty^6} \sum_{\boldsymbol{n}\in\mathbb{Z}^6} q^{\boldsymbol{n}^T C(E_6)\boldsymbol{n}} \\
&= 1 + 78q + 729q^2 + 4382q^3 + 19917q^4 + 77274q^5 + \cdots\ ,
\end{aligned}
\tag{2.39}
$$

which is the character of the affine $E_6$ algebra at level 1. Once again, the Sugawara central charge of $(E_6)_1$ is $c = 6$, which agrees with our expectation.

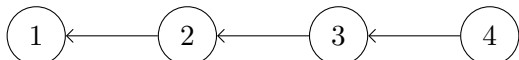

**Figure 3**: BPS quiver for the $(A_1, A_4)$ theory in the linear chamber

### 2.3 $(A_1, A_4)$ theory

In this section, we consider $(A_1, A_4)$ Argyres-Douglas theory, which is of rank-2. Its BPS quiver (in some particular chamber) is given in figure 3. The monodromy operator at this chamber is given as

$$M = \Psi_q(X_{\gamma_1})\Psi_q(X_{\gamma_2})\Psi_q(X_{\gamma_3})\Psi_q(X_{\gamma_4})\Psi_q(X_{-\gamma_1})\Psi_q(X_{-\gamma_2})\Psi_q(X_{-\gamma_3})\Psi_q(X_{-\gamma_4}) . \tag{2.40}$$

One can rewrite this operator using (2.5) to get

$$M = \Theta(X_{\gamma_1})\Psi_q(X_{\gamma_2-\gamma_1})\Theta(X_{\gamma_2})\Psi_q(X_{\gamma_3-\gamma_2})\Theta(X_{\gamma_3})\Psi_q(X_{\gamma_4-\gamma_3})\Theta(X_{\gamma_4}) . \tag{2.41}$$

The central charge for the VOA is given as

$$c_{2d}^{(N)} = \frac{40N}{7} - 4 , \tag{2.42}$$

which gives $c = \frac{12}{7}$ for $N = 1$ and $c = \frac{52}{7}$ for $N = 2$.

#### 2.3.1 $\mathrm{Tr}M$: $\mathrm{osp}(1|4)_1$

The trace of the monodromy operator can be evaluated straight-forwardly

$$
\begin{aligned}
I_1(q) &= (q)_\infty^4 \mathrm{Tr}M \\
&= \frac{1}{(q)_\infty^4} \sum_{\boldsymbol{n}\in\mathbb{Z}^4} \sum_{\boldsymbol{k}\in\mathbb{Z}^3_{\geq 0}} \frac{q^{\frac{1}{2}\boldsymbol{n}\cdot\boldsymbol{n}}}{(q)_{k_1}(q)_{k_2}(q)_{k_3}} \mathrm{Tr}X_{\gamma_1}^{n_1-k_1} X_{\gamma_2}^{k_1+n_2-k_2} X_{\gamma_3}^{k_2+n_3-k_3} X_{\gamma_4}^{k_3+n_4} \\
&= \frac{1}{(q)_\infty^4} \sum_{\boldsymbol{k}\in\mathbb{Z}^3_{\geq 0}} \frac{q^{\frac{1}{2}\boldsymbol{k}^T C(A_3)\boldsymbol{k}}}{(q)_{k_1}(q)_{k_2}(q)_{k_3}}
\end{aligned}
\tag{2.43}
$$

where $C(A_3)$ is the Cartan matrix for the $A_3$ Lie algebra. Upon expanding in $q$-series, we obtain

$$I_1(q) = 1 + 6q + 12q^2 + 28q^3 + 57q^4 + 108q^5 + 191q^6 + \cdots , \tag{2.44}$$

which was conjectured to be the specialized vacuum character of affine $\mathrm{osp}(1|4)$ algebra at level 1 in [14].

**2.3.2** $\mathrm{Tr}M^2$: $(X_1)_1$

Using the wall-crossing identities, we can organize $\mathrm{Tr}M^2$ into

$$
\mathrm{Tr}M^2 = \mathrm{Tr}\ \Psi_q(X_{-\gamma_4})\Psi_q(X_{-\gamma_2})\Theta(X_1)\Theta(X_2)\Theta(X_1)\Theta(X_3)\Theta(X_2) \\
\cdot\ \Theta(X_1)\Theta(X_4)\Theta(X_3)\Theta(X_2)\Theta(X_1) \tag{2.45}
$$

Evaluating the trace, we find

$$
I_2(q) = (q)_\infty^4 \mathrm{Tr}M^2
$$

$$
= \frac{1}{(q)_\infty^6} \sum_{n_1,n_2\geq 0} \sum_{(k_1,\cdots k_{10})\in\mathbb{Z}^{10}} \frac{q^{\frac{1}{2}(n_1^2+n_2^2+k_1^2+\cdots k_{10}^2)+k_1k_2-k_5k_6-k_9k_{10}-k_7k_8-k_5k_8-k_5k_{10}}}{(q)_{n_1}(q)_{n_2}} \tag{2.46}
$$

$$
\cdot\ \delta_{k_1+k_3+k_6+k_{10},0}\cdot\delta_{k_4+k_8,0}\cdot\delta_{n_1-k_7,0}\cdot\delta_{-n_2+k_2+k_5+k_9,0}\ ,
$$

which can also be written as

$$
I_2(q) = (q)_\infty^4 \mathrm{Tr}M^2 = \frac{1}{(q)_\infty^6} \sum_{n_1,n_7\geq 0} \frac{1}{(q)_{n_1}(q)_{n_7}} \sum_{\{n_2,\cdots n_6,n_8\}\in\mathbb{Z}^6} q^{\frac{1}{2}\mathbf{n}^t K_8' \mathbf{n}}\ , \tag{2.47}
$$

where we defined $\mathbf{n} = (n_1,\cdots,n_8)$ and

$$
K_8' = \begin{pmatrix}
2 & 0 & 0 & 0 & 0 & 1 & 0 & 0 \\
0 & 2 & 0 & 1 & 0 & 1 & -1 & -1 \\
0 & 0 & 2 & 0 & 1 & 0 & 1 & 1 \\
0 & 1 & 0 & 2 & -1 & 0 & -1 & -1 \\
0 & 0 & 1 & -1 & 2 & 0 & 1 & 1 \\
1 & 1 & 0 & 0 & 0 & 2 & 0 & 0 \\
0 & -1 & 1 & -1 & 1 & 0 & 2 & 1 \\
0 & -1 & 1 & -1 & 1 & 0 & 1 & 2
\end{pmatrix}. \tag{2.48}
$$

Again the matrix $K_8'$ defines a rank eight even positive-definite unimodular, which must be isomorphic to the $E_8$ lattice. One can construct a $SL(2,\mathbb{Z})$ matrix $U'$ such that $K_8' = U'C(E_8){U'}^T$, where

$$
U' = \begin{pmatrix}
1 & 0 & 0 & 0 & 0 & 0 & 0 & 0 \\
0 & 0 & 1 & 2 & 2 & 1 & 0 & 1 \\
0 & 0 & 0 & 0 & -1 & -1 & 0 & -1 \\
0 & 0 & 1 & 1 & 1 & 1 & 0 & 0 \\
0 & 0 & 0 & 0 & -1 & -1 & 0 & 0 \\
0 & -1 & -1 & 0 & 0 & 0 & 0 & 0 \\
0 & 0 & 0 & 0 & 0 & 0 & 1 & 0 \\
0 & 0 & -1 & -1 & -2 & -1 & 0 & -1
\end{pmatrix}. \tag{2.49}
$$

Therefore this basis change leaves the two vectors $\alpha_1$ and $\alpha_7$ invariant. This allows us to write the index in terms of $C(E_8)$:

$$
I_2(q) = (q)_\infty^4 \mathrm{Tr}M^2 = \frac{1}{(q)_\infty^6} \sum_{n_1,n_7\geq 0} \frac{1}{(q)_{n_1}(q)_{n_7}} \sum_{\{n_2,\cdots n_6,n_8\}\in\mathbb{Z}^6} q^{\frac{1}{2}\mathbf{n}^t C(E_8)\mathbf{n}}\ . \tag{2.50}
$$

Performing the series expansion, we find

$$I_2(q) = (q)^4_\infty \text{Tr} M^2 = 1 + 156q + 2236q^2 + 17056q^3 + \cdots , \tag{2.51}$$

The character in (2.51) was found as a solution to the third order modular differential equation in [45]. The connection to the intermediate algebra $X_1$ [46] was observed in [47], claiming that this gives the character of the affine $X_1$ algebra of level 1.

## 2.4 $(A_1, D_4)$ theory

The BPS quiver for the $(A_1, D_4)$ theory is given as in the figure 4. The monodromy operator for this theory is given as

$$M = \Psi_q(X_{\gamma_1})\Psi_q(X_{\gamma_3})\Psi_q(X_{\gamma_4})\Psi_q(X_{\gamma_2})\Psi_q(X_{-\gamma_1})\Psi_q(X_{-\gamma_3})\Psi_q(X_{-\gamma_4})\Psi_q(X_{-\gamma_2}) . \tag{2.52}$$

This theory has $SU(3)$ flavor symmetry, which is of rank 2. The flavor charges are given by

$$\gamma_{f_1} = \gamma_4 - \gamma_3 , \quad \gamma_{f_2} = \gamma_1 - \gamma_2 , \tag{2.53}$$

The central charge for the VOA is given as

$$c_{2d}^{(N)} = 6N - 2 . \tag{2.54}$$

### 2.4.1 $\text{Tr} M$: $(D_4)_1$

Using the wall-crossing identities, we find

$$\text{Tr} M = \Theta(X_{\gamma_1})\Theta(X_{\gamma_2})\Theta(X_{\gamma_3})\Theta(X_{\gamma_4})\Theta(X_{\gamma_2})\Theta(X_{\gamma_4}) \tag{2.55}$$

Evaluating the trace, we find

$$\begin{aligned}
I_1(q) = (q)^2_\infty \text{Tr} M &= \frac{1}{(q)^4_\infty} \sum_{(n_1,n_2,n_3,n_4)\in\mathbb{Z}^4} q^{\frac{1}{2}(n_1^2+2n_2^2+n_3^3+n_4^2+(n_1+n_2+n_4)^2)-n_2n_3-n_2n_4} \\
&= \frac{1}{(q)^4_\infty} \sum_{\mathbf{n}\in\mathbb{Z}^4} q^{\frac{1}{2}\mathbf{n}^t K_4 \mathbf{n}} \\
&= \frac{1}{(q)^4_\infty} \sum_{\mathbf{n}\in\mathbb{Z}^4} q^{\frac{1}{2}\mathbf{n}^t C(D_4)\mathbf{n}}
\end{aligned} \tag{2.56}$$

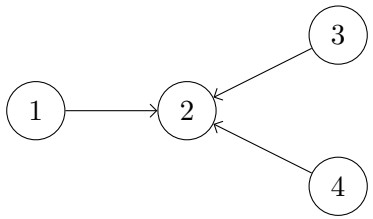

**Figure 4**: BPS quiver for the $(A_1, D_4)$ theory

where $C(D_4)$ is the Cartan matrix of $D_4$. The last equality comes from the existence of a $GL(4,\mathbb{Z})$ matrix

$$U = \begin{pmatrix} 1 & 1 & 0 & 1 \\ 0 & 1 & 0 & 0 \\ 0 & 1 & -1 & 0 \\ 0 & 0 & 0 & 1 \end{pmatrix}. \tag{2.57}$$

such that $U^t C(D_4) U = K_4$. This expression coincides with the vacuum character of the VOA $(D_4)_1$, which has the $q$-expansion

$$I_1(q) = (q)^2_\infty \operatorname{Tr} M = 1 + 28q + 134q^2 + 568q^3 + \cdots . \tag{2.58}$$

Once again, the Sugawara central charge for the $(D_4)_1$ is $c = 4$, agreeing with our expectation based on BPS monodromy.

## 3 Intermediate VOAs from twisted 3d SCFTs

The BPS monodromy trace allows us to write the putative character in the form of Nahm sum (or fermionic expression). This expression can be interpreted as a half-index of certain 3d $\mathcal{N} = 2$ Chern-Simons matter theory [25]. This 3d theory exhibits supersymmetry enhancement to $\mathcal{N} = 4$ theory along the renormalization group flow. Upon topological twisting, we land on a bulk topological field theory that has a boundary chiral theory whose partition function is given by the BPS monodromy trace.

In this section, we focus on two particular examples that are closely related to the so-called intermediate vertex subalgebra $(E_{7\frac{1}{2}})_1$ and $(X_1)_1$. We obatined such VOAs from $(A_1, A_2)$ and $(A_1, A_4)$ theory respectively from the BPS monodromy. Our construction provides a UV complete, consistent bulk topological field theory with boundary algebra whose vacuum character coincides with one of the modular invariant characters of the intermediate vertex algebras.

### 3.1 "$(E_{7\frac{1}{2}})_1$"

#### 3.1.1 An $\mathcal{N} = 2$ Lagrangian description

We start from an observation that the Nahm sum formula (2.27) can be thought of as a half-index of a simple 3d $\mathcal{N} = 2$ Chern-Simons matter theory,

$$\mathcal{N} = 2 \quad U(1)^8_K \ + \ 1 \ \text{ chiral multiplet} , \tag{3.1}$$

where the chiral multiplet has charge 1 under the first $U(1)$ factor of the gauge group. The (effective) mixed Chern-Simons level $K$ is identified with the Cartan matrix of $E_8$, $C(E_8)$. This theory can be represented as a quiver diagram depicted in figure 5. where each of the solid lines corresponds to a mixed CS interaction with level $-1$, and each of the gauge group factors has the effective CS level 2.

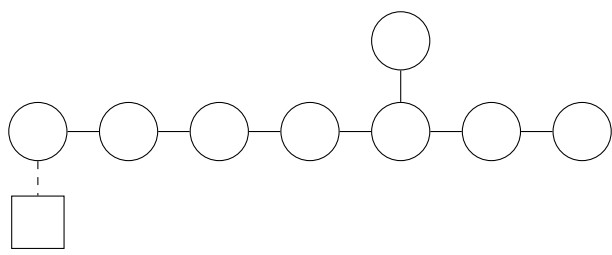

**Figure 5**: 'Quiver diagram' for the "$E_{7\frac{1}{2}}$" theory. Each solid line corresponds to a mixed CS interaction with level $-1$ (not the bifundamental chiral multiplet), and each of the gauge group factors has the effective CS level 2. The dashed line corresponds to the chiral multiplet whose charge is 1.

The gauge theory description (3.1) has a global symmetry $U(1)_R \times U(1)_A$, where the factor $U(1)_A$ is the flavor symmetry that corresponds to a particular linear combination of the $U(1)$ topological symmetries of the eight $U(1)$ gauge group factors. All the other topological symmetries decouple in the infrared, since there is no gauge-invariant operator charged under such symmetries. It can also be read from the superconformal index we discuss below. In the following, we provide evidence that the theory flows to a superconformal theory with $\mathcal{N} = 4$ supersymmetry enhancement.

### 3.1.2 Supersymmetry enhancement

It is convenient to identify the $U(1)_A$ global symmetry with

$$U(1)_A = U(1)_{\alpha_1} := 2U(1)_1^{\text{top}} - U(1)_2^{\text{top}} \ , \tag{3.2}$$

which is a linear combination of the topological symmetries for the first two $U(1)$ gauge group factors. Let us consider a twisted R-symmetry

$$R_\nu = R_0 + \nu A \ , \tag{3.3}$$

with $\nu \in \mathbb{R}$, where $R_0$ is the reference R-charge which can be chosen to be the superconformal R-charge. We perform F-maximization [48] to fix the superconformal R-charge in the infrared.

It is useful to compute the superconformal index to test supersymmetry enhancement. The index is defined as

$$I(q, \eta; \nu) = \text{Tr}(-1)^{R_\nu} q^{\frac{R_\nu}{2} + j_3} \eta^A \ , \tag{3.4}$$

which can be computed via supersymmetric localization [49, 50]. We find that the superconformal index at the IR fixed point has an expansion

$$I(q, \eta; \nu = 0) = 1 - q - \left(\eta + \eta^{-1}\right) q^{3/2} - 2q^2 - \left(\eta + \eta^{-1}\right) q^{5/2} - 2q^3 - \cdots \ , \tag{3.5}$$

where $\eta$ is the $U(1)_A$ fugacity. The existence of the term $-\left(\eta + \eta^{-1}\right) q^{3/2}$ signals the presence of the extra supercurrent multiplet in the $\mathcal{N} = 4$ superconformal algebra [51, 52].[7] Furthermore, this expression coincides with the superconformal index of the minimal $\mathcal{N} = 4$ rank-zero theory $\mathcal{T}_{\min}$, which was first considered in [30]. This signals that our $U(1)_K^8$ gauge theory is (almost) dual to the $\mathcal{T}_{\min}$ as a CFT, but realizing a distinct boundary VOA! This suggests that the local operator spectrum of $\mathcal{T}_{\min}$ is identical to our gauge theory in the infrared, but the 'global structure' or 'topological degrees of freedom' can differ. To understand this, let us take a closer look at the topological sector of two theories by computing more general partition functions.

The low-energy $\mathcal{N} = 4$ theory can be topologically twisted in two different ways to produce two distinct TFTs, which correspond to the choice $\nu = \pm 1$. The twisted partition function on a circle bundle over a genus-$g$ Riemann surface can be computed [53–57] to extract a part of the modular data of the boundary algebra,

$$|S_{0\alpha}| \quad \text{and} \quad T_{\alpha\alpha} , \tag{3.6}$$

up to an overall phase. (See e.g., [32] for a related discussion.) For the $\nu = 1$ twist, we find that the result is compatible with the following modular representation (up to an overall phase factor that we do not keep track of):

$$S = \frac{2}{\sqrt{5}} \begin{pmatrix} \sin(2\pi/5) & \sin(\pi/5) \\ \sin(\pi/5) & -\sin(2\pi/5) \end{pmatrix} , \quad T = \mathrm{diag}\left(e^{2\pi i(-19/60)}, e^{2\pi i(29/60)}\right) , \tag{3.7}$$

which reproduces the modular data of $(\hat{E}_{7\frac{1}{2}})_1$ [17]. Note that the same data, (3.6), obtained from the bulk theory is also compatible with that of the Lee-Yang minimal model $M(2,5)$, whose modular representation can be written as

$$S_{LY} = \frac{2}{\sqrt{5}} \begin{pmatrix} -\sin(2\pi/5) & \sin(\pi/5) \\ \sin(\pi/5) & \sin(2\pi/5) \end{pmatrix} , \quad T_{LY} = \mathrm{diag}\left(e^{2\pi i(11/60)}, e^{2\pi i(-1/60)}\right) . \tag{3.8}$$

These two modular data produce the same partition functions on a large class of Seifert manifolds up to an overall phase, which we do not keep track of in the twisted partition function computation.

Our analysis gives strong evidence that the IR fixed point of the theory (3.1) is dual to $\mathcal{T}_{\min}$, possibly up to a multiplication of an invertible topological field theory (TFT).

### 3.1.3 Other modules

We conjecture that the Wilson line $W_1$ of gauge charge $(2, -1, 0, \cdots, 0)$ flows to the line operator in the twisted topological field theory, which corresponds to another module whose

---

[7]There exist two gauge invariant 1/4-BPS dressed monopole operators in the theory, $\phi_1 V_{\pm\theta}$, which can contribute to this term. Here $V_m$ is a bare monopole operator with the flux vector $m$, and $\theta = (2, 3, 4, 5, 6, 4, 2, 3)$ is the highest root of the $E_8$ algebra.

character forms a vector-valued modular function with (2.28). Let $\mathbf{x} = (x_1, \cdots, x_8)$ be the gauge holonomy variables for the UV gauge theory. One can check that the following relation holds

$$\langle W_\alpha \rangle_\beta = L^\alpha(\mathbf{x}^{(\beta)}) = \frac{S_{\alpha\beta}}{S_{0\beta}} \ , \tag{3.9}$$

for $L^0 = 1$ and $L^1 = x_1^2 x_2^{-1}$, where $\mathbf{x}^{(0)}$ and $\mathbf{x}^{(1)}$ are the two solutions to the Bethe equation of the original gauge theory, which is a system of 8 polynomial equations in $\mathbf{x} = \{x_a = e^{2\pi i u_a}\}$,

$$P_a(\mathbf{x}) = \exp\left[2\pi i \frac{\partial W(u)}{\partial u_a}\right] = 1 \ , \tag{3.10}$$

where $W$ is the Coulomb branch twisted effective superpotential of the gauge theory. See, e.g., [32] for more details.

### 3.1.4 Boundary condition

For the $\nu = 1$ twist, we consider the supersymmetric Dirichlet boundary condition for all of the $\mathcal{N} = 2$ vector and chiral multiplets [35]. The half-index with this boundary condition reads

$$I_{\text{half}}(q; s_1, \cdots s_8) = \frac{1}{(q)_\infty^8} \sum_{\mathbf{n}\in\mathbb{Z}^8} q^{\frac{1}{2}\mathbf{n}^t C(E_8)\mathbf{n}} (q^{1-n_1} s_1^{-1}; q)_\infty \prod_{i,j=1}^{8} s_i^{C(E_8)_{ij} n_j} \ , \tag{3.11}$$

where $s_i$'s are fugacity for the boundary currents $U(1)_\partial^8$. Specializing $s_i \to 1$, we obtain

$$I_{\text{half}}(q; 1, \cdots, 1) = \frac{1}{(q)_\infty^7} \sum_{n_1 \geq 0} \sum_{\{n_2, \cdots, n_8\}\in\mathbb{Z}^7} \frac{q^{\frac{1}{2}\mathbf{n}^t C(E_8)\mathbf{n}}}{(q)_{n_1}} \ , \tag{3.12}$$

which coincides with (2.27).

The non-vacuum character of the boundary algebra can be obtained by inserting the simple line $W_1$. Specializing to the limit $s_i \to 1$, we find

$$\begin{aligned}
I_{\text{half}}[W_1](q; 1, \cdots, 1) &= \frac{1}{(q)_\infty^7} \sum_{n_1 \geq 0} \sum_{\{n_2, \cdots, n_8\}\in\mathbb{Z}^7} \frac{q^{\frac{1}{2}\mathbf{n}^t C(E_8)\mathbf{n} + 2n_1 - n_2}}{(q)_{n_1}} \\
&= \frac{1}{(q)_\infty^7} \sum_{n_1 \geq 1} \sum_{\{n_2, \cdots, n_8\}\in\mathbb{Z}^7} \frac{q^{\frac{1}{2}\mathbf{n}^t C(E_8)\mathbf{n}}}{(q)_{n_1-1}} \ , \\
&= 57 + 1102q + 9367q^2 + 57362q^3 + \cdots \ ,
\end{aligned} \tag{3.13}$$

which is another modular invariant character of $(E_{7\frac{1}{2}})_1$.

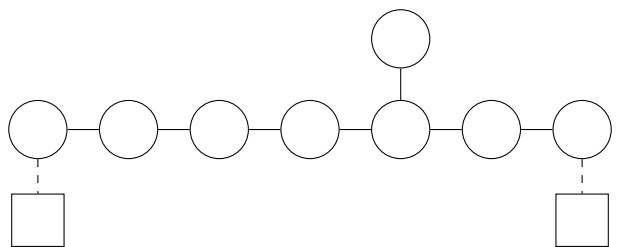

**Figure 6**: 'Quiver diagram' for the "$X_1$" theory. Each solid line corresponds to a mixed CS interaction with level $-1$, and each of the gauge group factors has the effective CS level 2. The dashed line corresponds to the chiral multiplet whose charge is 1.

### 3.2 "$(X_1)_1$"

#### 3.2.1 An $\mathcal{N} = 2$ Lagrangian description

The Nahm sum type formula for (2.50) suggests a 3d $\mathcal{N} = 2$ Chern-Simons matter theory description

$$\mathcal{N} = 2 \;\; U(1)_K^8 \;\; + \;\; 2 \text{ chiral multiplets} , \tag{3.14}$$

where the chiral multiplets has charge 1 under the first $U(1)$ factor and the seventh $U(1)$ factor of the gauge group. The (effective) mixed Chern-Simons level $K$ is identified with the Cartan matrix of $E_8$, $C(E_8)$. This theory can be represented as a quiver diagram depicted in figure 6. This theory is almost identical to the one discussed in section 3.1, except that we added one extra charged chiral multiplet.

The gauge theory description allows a half-BPS monopole operator of the form:

$$V_{(2,2,2,2,2,1,0,1)}\phi_7 , \tag{3.15}$$

where $V_{\mathbf{m}}$ is a bare monopole operator with gauge flux $\mathbf{m}$ and $\phi_7$ is the scalar field in the second chiral multiplet (attached to the seventh node). Deforming the theory with the monopole superpotential, the global symmetry that acts non-trivially on the IR theory is $U(1)_A \times U(1)_R$, as in the previous example.

#### 3.2.2 Supersymmetry enhancement

We identify the $U(1)_A$ symmetry as

$$U(1)_A = U(1)_1^{\text{top}} - U(1)_2^{\text{top}} + U(1)_7^{\text{top}} . \tag{3.16}$$

The superconformal index at the IR fixed point can be computed to give

$$I(q, \eta; \nu = 0) = 1 - q - \left(\eta + \eta^{-1}\right) q^{3/2} - 2q^2 - \eta q^{5/2} + \left(-1 + \frac{1}{\eta^2}\right) q^3 + \cdots , \tag{3.17}$$

where again $\eta$ is the $U(1)_A$ fugacity. One can check the first several coefficients of the expansion coincide with the superconformal index of the $\mathcal{N} = 4$ rank-zero SCFT called $\mathcal{T}_2$,

which was considered first in [32]. This suggests that our theory is (almost) dual to the $\mathcal{T}_2$ theory.

The low-energy $\mathcal{N} = 4$ superconformal theory can be topologically twisted to produce two distinct TFTs, which corresponds to the choice $\nu = \pm 1$, where $\nu$ is a twisting parameter we introduced in (3.3). For the $\nu = 1$ twist, the modular data (3.6) extracted from the partition function on Seifert manifolds are compatible with the following representation:

$$S = \frac{2\sin(\pi/7)}{\sqrt{7}} \begin{pmatrix} d^2-1 & d & 1 \\ d & -1 & 1-d^2 \\ 1 & 1-d^2 & d \end{pmatrix} , \tag{3.18}$$

where $d = 2\cos(\pi/7)$ , and

$$T = e^{2\pi i(-13/42)} \mathrm{diag}\left(1, e^{2\pi i(6/7)}, e^{2\pi i(4/7)}\right) . \tag{3.19}$$

We can check that this modular data is almost, but not quite identical to, that of the Virasoro minimal model $M(2,7)$, which can be obtained via topologically twisted $\mathcal{T}_2$. Each component of modular data differs by a phase. Therefore, we claim that this theory is dual to $\mathcal{T}_2$ up to a possible invertible topological field theory.

### 3.2.3 Other modules

We conjecture that the following two Wilson lines

$$W_1 = (1, -1, 0, \cdots, 0, 1, 0) , \quad W_2 = (0, -1, 0, \cdots, 0, 1) , \tag{3.20}$$

flows to the line operators in the twisted topological field theory, which corresponds to the two modules that transform as a vector-valued modular form together with (2.51). Let $\mathbf{x} = (x_1, \cdots, x_8)$ be the gauge holonomy variables for the UV gauge theory. One can check that the following relation holds

$$\langle W_\alpha \rangle_\beta = L^\alpha(\mathbf{x}^{(\beta)}) = \frac{S_{\alpha\beta}}{S_{0\beta}} , \tag{3.21}$$

for $L^0 = 1$, $L^1 = x_1^1 x_2^{-1} x_7$, and $L^2 = x_2^{-1} x_8$, where $\{\mathbf{x}^{(0)}\}$, $\{\mathbf{x}^{(1)}\}$, and $\{\mathbf{x}^{(2)}\}$ are the three solutions to the Bethe equation of the gauge theory.

### 3.2.4 Boundary condition

For the $\nu = 1$ twist, imposing the supersymmetric Dirichlet boundary conditions for all of the $\mathcal{N} = 2$ vector and chiral multiplets gives the half-index

$$I_{\mathrm{half}}(q; s_1, \cdots s_8) = \frac{1}{(q)_\infty^8} \sum_{\mathbf{n} \in \mathbb{Z}^8} q^{\frac{1}{2}\mathbf{n}^t C(E_8)\mathbf{n}} (q^{1-n_1} s_1^{-1}; q)_\infty (q^{1-n_7} s_7^{-1}; q)_\infty \prod_{i,j=1}^{8} s_i^{C(E_8)_{ij} n_j} \tag{3.22}$$

where again $s_i$'s are the fugacities for boundary $U(1)_\partial^8$ currents. Specializing to $s_i \to 1$, we recover the vacuum character of the affine version of the $X_1$ algebra [46] at level 1:

$$I_{\text{half}}(q; 1, \cdots, 1) = \sum_{n_1, n_7 \geq 0} \frac{1}{(q)_{n_1} (q)_{n_7}} \sum_{\{n_2, \cdots n_6, n_8\} \in \mathbb{Z}^6} q^{\frac{1}{2} \mathbf{n}^t C(E_8) \mathbf{n}} . \tag{3.23}$$

The two other modular invariant characters of $(X_1)_1$ algebra can be obtained by inserting the two line operators (3.20), which gives

$$\begin{aligned} I_{\text{half}}[W_1](q; 1, \cdots, 1) &= \sum_{n_1, n_7 \geq 0} \frac{1}{(q)_{n_1} (q)_{n_7}} \sum_{\{n_2, \cdots n_6, n_8\} \in \mathbb{Z}^6} q^{\frac{1}{2} \mathbf{n}^t C(E_8) \mathbf{n}} q^{n_1 - n_2 + n_7} \\ &= 13 + 364q + 3302q^2 + \cdots , \end{aligned} \tag{3.24}$$

and

$$\begin{aligned} I_{\text{half}}[W_2](q; 1, \cdots, 1) &= \sum_{n_1, n_7 \geq 0} \frac{1}{(q)_{n_1} (q)_{n_7}} \sum_{\{n_2, \cdots n_6, n_8\} \in \mathbb{Z}^6} q^{\frac{1}{2} \mathbf{n}^t C(E_8) \mathbf{n}} q^{-n_2 + n_8} \\ &= 78 + 1288q + 10465q^2 + \cdots , \end{aligned} \tag{3.25}$$

which coincide with the $q$-expansion of $\chi_{4/7}$ and $\chi_{6/7}$ of affine $X_1$ at level one discussed in [47], which appears as solutions to the third order modular linear differential equation [45].

## 4 Discussion

In this paper, we have discovered a correspondence between multiple powers of the BPS monodromy operator for a given 4d $\mathcal{N} = 2$ SCFT and a family of vertex operator algebras (VOAs). This connection can be understood via $U(1)_r$-twisted compactification of the 4d theory, which gives rise to a 3d $\mathcal{N} = 4$ SCFT. Upon suitable topological twisting of the 3d theory and by putting the theory on a space with a boundary, we obtain corresponding boundary VOA. The family of VOAs arises from the multiplicity of twisting as we reduce the 4d theory to 3d.

We have found various (rational) VOAs realized from the BPS monodromy of Argyres-Douglas theories, especially a series of VOAs given by affine Lie algebras in the Deligne-Cvitanović exceptional series of level 1. Remarkably, we are able to find the characters of the 'intermediate vertex subalgebras' such as $E_{7\frac{1}{2}}$ and $X_1$. It has been conjectured that 3d $\mathcal{N} = 4$ rank-0 theories give rise to rational VOAs [13, 24, 31, 32]. However, it is not known whether there can be a rational VOA whose vacuum character is given by the modular-invariant character of affine $E_{7\frac{1}{2}}$ and $X_1$ at level 1. Note that the characters we obtain also appear in a twisted sector of another algebra. For example, $(E_{7\frac{1}{2}})_1$ characters can be obtained from the W-algebra $W_{-5}(E_8, f_\theta)$ which is rational [42].

The 3d $\mathcal{N} = 4$ theories associated to the $E_{7\frac{1}{2}}$ and $X_1$ have another interesting property. Namely, their superconformal indices and the twisted partition functions (on a circle fibration over a Riemann surface) agree with that of $\mathcal{T}_{\min} = \mathcal{T}_1$ and $\mathcal{T}_2$ theories of [30, 31] that gives

Virasoro minimal models as boundary VOAs, up to an overall phase. It hints that our gauge theories are IR dual to $\mathcal{T}_1$ and $\mathcal{T}_2$, up to an invertible TQFT. It would be interesting to explicitly compute the boundary OPEs of these 3d topological field theories to test the duality further. We leave it as a future work.

In some sense, our construction provides a generalization of the SCFT/VOA correspondence of [3], which comes from $\text{Tr} M^{-1}$ in our language. However, unlike the case of [3], where the associated VOA can be defined purely in terms of 4d superconformal field theory data by passing to certain cohomology, we do not have such a direct construction. It would be interesting to clarify the connection between other VOAs coming from $\text{Tr} M^N$ and 4d physics at the conformal point.

There are numerous questions we can ask regarding our connection between 4d $\mathcal{N} = 2$ SCFT, 3d $\mathcal{N} = 4$ SCFT, and VOAs. We would like to conclude by making some comments for the future directions.

- Connection to the holomorphic modular bootstrap program [15, 16]: We find that for a given AD theory, we get a family of VOAs whose characters solve the MLDEs of the same degree [45, 58, 59]. Why is this true? It is conjectured that for any 4d $\mathcal{N} = 2$ SCFT, the Schur index should obey a finite order MLDE [60]. Is there any connection between the two?

- Other powers of monodromy operator: For Argyres-Douglas theories, there exists a positive integer $n$, such that $n$-th wrapping of the Janus configuration trivializes the $U(1)_r$-twisting [8]. Therefore, it is natural to expect that there exists some sort of periodicity in the higher powers of monodromy traces. However, it appears that $\text{Tr} M^n$ does not give rise to a converging $q$-series. It would be interesting to see if there is a way to systematically regularize this sum as was done in [61].

- Fusions and Modular properties of VOAs: Characters for other simple modules can be obtained via inserting surface defects [62, 63]. The trace formula for the $S^3$-partition functions and the modular data can be obtained for the higher powers of monodromy as well [25], which will be discussed in an upcoming work [64].

- Higher-rank AD theories: It would be interesting to further investigate the correspondence to higher-rank Argyres-Douglas theories. For the $(A_1, G)$ type AD theories, the BPS quiver is known and we can straightforwardly compute the monodromy traces. For other $(G, G')$ theories, the corresponding BPS quiver does not encode the full BPS spectrum. Therefore it is not known how to compute the monodromy operator. Instead, one can use $\mathcal{N} = 1$ Lagrangian description [65, 66] for certain cases of AD theories and follow the strategy of [26, 27]. However, this method becomes computationally demanding for higher powers of monodromy (or higher sheets) and higher rank.

- Non-Argyres-Douglas theories: In our study, it was crucial to have a Coulomb branch operator with fractional $U(1)_r$ charge to have a non-trial twisting. However, nothing

stops us from computing the higher powers of the monodromy operator for arbitrary $\mathcal{N} = 2$ SCFTs. It would be interesting to ask if it gives us any interesting VOAs.

- VOA-valued TQFTs from class $\mathcal{S}$: It is known that one can construct a vertex operator algebra-valued TQFT [67, 68] by considering class $\mathcal{S}$ theories [69, 70]. Here VOA is associated with $\text{Tr}M^{-1}$ of the corresponding class $\mathcal{S}$ theory. It is interesting to ask if there can be a TQFT structure for the VOAs constructed from $\text{Tr}M^N$ of class $\mathcal{S}$ theory. The class $\mathcal{S}$ also included a large set of Argyres-Douglas theories [71, 72], whose associated VOA is particularly simple [73, 74]. It would be interesting to ask if such a structure persists.

## Acknowledgments

We thank T. Arakawa, D. Gaiotto, and D. Gang for useful comments and discussions. The work of HK is supported by the National Research Foundation of Korea (NRF) grant NRF-2023R1A2C1004965. The work of JS is supported by the National Research Foundation of Korea (NRF) grant RS-2023-00208602 and also by the POSCO Science Fellowship of POSCO TJ Park Foundation. This work is also supported by the National Research Foundation of Korea (NRF) grant RS-2024-00405629.

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
