# Peer review of "A Family of Vertex Algebras from Argyres-Douglas Theory"

_SciPost Physics, doi:SciPost Phys. 19, 144 (2025)_

## Round 2 · Referee Report · Anonymous (Referee 1) · 2025-7-15

Report

This paper should eventually be published in this journal after minor modifications are made in response to my comments.

This is an interesting paper that takes forward earlier ideas on the relation between 4d N=2 SCFTs and 2d RCFTs. Cordova-Shao had proposed that the trace of the inverse monodromy matrix M (times a prefactor coming from free vector multiplets) gives the Schur index of the 4d theory. Subsequently Cecotti-Song-Vafa-Yan generalised these ideas to consider traces of positive powers $M^N$ and tried to identify them with 2d CFTs. Despite a few successes, these identifications seemed (at least to me) somewhat conjectural and, at times, artificial. The claim of the present work is that instead of choosing an N-dependent prefactor as in Cecotti et al, one should retain the original prefactor for the Schur index and only change the power of the matrix M. This leads the authors to the vacuum characters of some very well-known 2d CFTs (and a few exotic but mathematically well-understood ones associated to IVAs). In some cases the authors are able to precisely extract this vacuum character, while in other cases like G2 and F4 they make their identification based just on the leading terms. However even this is quite convincing. Overall this approach seems quite promising and there are very likely more examples on these lines.

Section 3 raises a puzzle. It is known that the c=38/5 theory has formally negative fusion rules following from the Verlinde formula. For this reason it is called an IVA and these are not considered standard CFTs. Yet, the authors claim to have found a “UV complete, consistent bulk TFT” which realises these IVAs as a boundary algebra. The puzzle is, if the TFT is fully consistent then why are the boundary theories not proper CFTs? Conversely, should one not expect that the bulk TFT also suffers from some inconsistency associated to the "negative" fusion rules? It would be good to say a few words about this.

Requested changes

Suggestions to improve the manuscript:

(i) It will be helpful to the reader to recall that $\langle \gamma_1,\gamma_2\rangle$ is an integral anti-symmetric pairing on the charge lattice.

(ii) Below Eq (1.1), the definition of the $q$-Pochhammer symbol needs to be slightly corrected – as it stands it vanishes, because the first term (for $n=0$) is 0. Either $n$ should start from 1, or the power should be $q^{n+1}$.

(iii) Para before Eq (1.2): “is considered” -> “are considered”.

(iv) Below Eq (2.9) “respectivey” -> “respectively”.

(v) Page 16, last para: “obatined” -> “obtained”.

Recommendation

Ask for minor revision

  • validity: -
  • significance: -
  • originality: -
  • clarity: -
  • formatting: -
  • grammar: -

Author:  Heeyeon Kim  on 2025-10-01  [id 5877]

(in reply to Report 1 on 2025-07-15)

We would like to thank the referee for carefully reading our manuscript. To address the concerns raised:

Section 3 is meant primarily as observations of the following facts rather than a definitive claim. 1) There is strong evidence suggesting the existence of an N=2 gauge theory that flows to an N=4 fixed point with zero-dimensional Coulomb and Higgs branches. Such a theory can be topologically twisted to produce TFTs, and one can attempt to extract modular data from partition function calculations in the UV. 2) In addition, if the Dirichlet boundary condition flows in the IR to a deformable boundary condition compatible with the twisted supercharge Q, then half-index computations can be used to probe the vacuum characters of the corresponding boundary algebras. In the examples in section 3, these computations lead to the characters of the so-called intermediate VAs.

As far as we understand, it is possible to define these IVAs as a consistent vertex algebra, even though it lies far outside the class of rational vertex operator algebras.

We recognize that these observations seem to be in some tension with the general expectation that twisted rank-zero theories give rise to rational boundary VOAs. On the other hand, identification of the boundary VOAs of these non-Lagrangian TFTs relies on assumptions about the IR SCFT and boundary conditions, which need to be carefully examined. We plan to revisit these issues in more detail in future work. In the meantime, in a revised version of the manuscript, we are planning to highlight the issue more explicitly and emphasize the distinction between observations, underlying assumptions and open questions.

---

## Round 2 · Referee Report · Anonymous (Referee 2) · 2025-9-26

Report

This paper investigates the interpretation of traces of powers of the BPS monodromy operator for four-dimensional N=2 SCFTs in terms of vertex operator algebra characters, generalising the infrared formula of Cordova-Shao for the Schur index. While some motivation for such an interpretation is given in terms of twisted compactifications of the corresponding theory in the presence of certain Janus configurations, such a derivation seems to be largely speculative, so concrete calculations to support the overall picture are welcome.

In the first section of the body of this paper, careful calculations are performed for a number of powers (N) for several Argyres-Douglas SCFTs, and proposed identifications with VOA characters are made. While an earlier work by Cecotti-Song-Vafa-Yan (CSVY) pursued a similar idea, here a different choice of prefactors for the traces is made. Consequently, while the identification of vertex algebras in CSVY seemed somewhat uncertain, the results in the present paper seem more compelling. With the given choice of prefactor, the authors recover (in some cases conjecturally, and in some cases provably) the characters of several well-known VOAs from the unitary Deligne-Cvitanovic series at level one, as well as two cases which are identified as "intermediate vertex algebras", which are more unconventional (in particular, they are not simple).

These two cases are of particular interest to the authors of this paper in the next section. The appearance of such objects in connection with the Deligne-Cvitanovic series is not new, but as vertex algebras these are much less well-behaved. In the paper, the authors use the form of the characters arising from their computational method to engineer three-dimensional N=2 Abelian Chern-Simons matter theories that they argue will flow to N=4 theories in the infrared whose topological twists admit these intermediate vertex alebras as boundary algebras.

The particular question of whether it is these intermediate vertex algebras that will be realised on the boundary of the given theories seems to me to be not completely satisfactorily addressed in the paper. As the authors point out, these vertex algebras are not rational and the fusion rules one would derive by taking seriously their characters (and those related by modular transformations) are not sensible. So the proposal that their three-dimensional theory, which is argued to flow in the IR to the same famous T_min theory that allows the rational (2,5) Virasoro VOA as a boundary chiral algebra, will admit these intermediate vertex algebras as boundary algebras seems hard to square with general expectations, and I'm not sure the computations supplied really are enough to substantiate the claim. I think that the paper would be improved either by strengthening the case for these claims with further calculations, or alternatively weakening the claims to allow for some uncertainty in the identification.

Recommendation

Ask for minor revision

  • validity: high
  • significance: good
  • originality: good
  • clarity: high
  • formatting: excellent
  • grammar: excellent

Author:  Heeyeon Kim  on 2025-10-01  [id 5878]

(in reply to Report 2 on 2025-09-26)

We would like to thank the referee for carefully reading our manuscript. To address the concerns raised:

Section 3 is meant primarily as observations of the following facts rather than a definitive claim. 1) There is strong evidence suggesting the existence of an N=2 gauge theory that flows to an N=4 fixed point with zero-dimensional Coulomb and Higgs branches. Such a theory can be topologically twisted to produce TFTs, and one can attempt to extract modular data from partition function calculations in the UV. 2) In addition, if the Dirichlet boundary condition flows in the IR to a deformable boundary condition compatible with the twisted supercharge Q, then half-index computations can be used to probe the vacuum characters of the corresponding boundary algebras. In the examples in section 3, these computations lead to the characters of the so-called intermediate VAs.

As far as we understand, it is possible to define these IVAs as a consistent vertex algebra, even though it lies far outside the class of rational vertex operator algebras.

We recognize that these observations seem to be in some tension with the general expectation that twisted rank-zero theories give rise to rational boundary VOAs. On the other hand, identification of the boundary VOAs of these non-Lagrangian TFTs relies on assumptions about the IR SCFT and boundary conditions, which need to be carefully examined. We plan to revisit these issues in more detail in future work. In the meantime, in a revised version of the manuscript, we are planning to highlight the issue more explicitly and emphasize the distinction between observations, underlying assumptions and open questions.

---

## Round 2 · Referee Report · Anonymous (Referee 3) · 2025-9-29

Report

It's a useful paper that examines characters of the higher-sheet boundary VOAs introduced recently in references [31,32], now through the lens of BPS monodromy operator. More specifically, 4D Argyres-Douglas theories, subject to the R-twisted circle reduction, flow to certain 3D theories, either conformal or gapped, whose boundary conditions support sensible VOAs. The simplest and best understood example is the $(A_1,A_2)$ AD theory leading to the minimal rank-0 SCFT and its gapped cousins, which yield the Galois orbit of the Lee-Yang MTC, which was recently thoroughly examined in the literature. The authors reproduce this orbit using the trace of the BPS monodromy method, and also perform similar analyses in other examples. In Section 3, they also look at the intermediate vertex algebras, including $(E_{7\frac12})_1$ in the $(A_1, A_2)$ case. In that case, they reproduce the expected characters for explicitly chosen boundary conditions, thus partially addressing the question asked in the end of Section 3.4 in [32]. It would also be very interesting to explicitly identify the generators of $(E_{7\frac12})_1$ as boundary operators in a 3D QFT.

I will go outside the scope of this report and also note that I find the comment made in the Referee#1's report a little bit surprising. They note that the intermediate algebra $(E_{7\frac12})_1$ has formally negative fusion rules. As far as I know, this algebra has Fibonacci-like fusion rules $\phi\times\phi = I + \phi$, see 1305.6463 and paragraph above eqn. (5.23) in 1912.11955.

Recommendation

Publish (meets expectations and criteria for this Journal)

---

## Round 4 · List of Changes

- We rewrote the first paragraph of Section 3 and the first paragraph of Section 3.1.3 to emphasize that the identification of the boundary VOA for these non-Lagrangian TFTs relies on assumptions about the IR SCFT and more importantly the deformability of the associated boundary conditions, which require careful examination.
- We also added a new paragraph on page 23, beginning with “The UV boundary condition…,” which reiterates this open problem and notes that it will be addressed in future work.
- A new reference [64] has been added.
- To better reflect the subtle and intricate nature of the intermediate algebras, we replaced “Vertex Operator Algebra” with “Vertex Algebra” in several places, including the title and the abstract.
- The paragraph surrounding Eq. (1.2) has been slightly revised.
- Minor typographical errors have been corrected.

---

## Editorial Decision

published